# A SPARSE, FAST, AND STABLE REPRESENTATION FOR MULTIPARAMETER TOPOLOGICAL DATA ANALYSIS

## ABSTRACT

Topological data analysis (TDA) is a new area of geometric data analysis that focuses on using invariants from algebraic topology to provide multiscale shape descriptors for point clouds. One of the most important shape descriptors is *persistent homology*, which studies the topological variations as a filtration parameter changes; a typical parameter is the feature scale.

For many data sets, it is useful to consider varying multiple filtration parameters at once, for example scale and density. While the theoretical properties of single parameter persistent homology are well understood, less is known about the multiparameter case. Of particular interest is the problem of representing multiparameter persistent homology by elements of a vector space for integration with traditional machine learning.

Existing approaches to this problem either ignore most of the multiparameter information to reduce to the one-parameter case or are heuristic and potentially unstable in the face of noise. In this article, we introduce a general representation framework for multiparameter persistent homology that encompasses previous approaches. We establish theoretical stability guarantees under this framework as well as efficient algorithms for practical computation, making this framework an applicable and versatile tool for TDA practitioners. We validate our stability results and algorithms with numerical experiments that demonstrate statistical convergence, prediction accuracy, and fast running times on several real data sets.

## 1 INTRODUCTION

Topological Data Analysis (TDA) (Carlsson, 2009) is a methodology for analyzing data sets using multiscale shape descriptors coming from algebraic topology. There has been intense interest in the field in the last decade, since topological features have allowed practitioners to compute and encode information that classical approaches do not capture. Moreover, TDA rests on solid theoretical grounds, with guarantees accompanying many of its methods and descriptors. TDA has proved useful in a wide variety of application areas, including computer graphics (Carrière et al., 2015a; Poulenard et al., 2018), computational biology (Rabadán & Blumberg, 2019), and material science (Buchet et al., 2018; Saadatfar et al., 2017), among many others.

The main tool of TDA is *persistent homology*. In its most standard form, one is given a finite metric space (e.g., a finite set of points and their pairwise distances) and a continuous function $f : X \to \mathbb{R}$. This function usually represents a parameter of interest (such as, e.g., scale or density for point clouds, marker genes for single-cell data, etc), and the goal of persistent homology is to characterize the topological variations of this function on the data. Of course, the idea of considering multiscale representations of geometric data is not new (Chapelle et al., 2002; Ozer, 2019; Witkin, 1987); the contribution of persistent homology is to obtain a novel and theoretically tractable multiscale shape descriptor. More formally, this is achieved by computing the so-called *persistence diagram* of $f$, which is obtained by looking at all sublevel sets of the form $\{f^{-1}((-\infty, \alpha])\}_\alpha$ and by computing a *decomposition* of these sets, that is, by recording the appearances and disappearances of topological features (connected components, loops, enclosed spheres, etc) in these sets. When such a feature appears (resp. disappears), e.g., in a sublevel set $f^{-1}((-\infty, \alpha_b])$, we call the corresponding threshold $\alpha_b$ (resp. $\alpha_d$) the *birth time* (resp. *death time*) of the topological feature, and we summarize this information in the persistence diagram $D(f) := \{(\alpha_b, \alpha_d)\}_{\alpha \in A} \subset \mathbb{R}^2$. Moreover, it is also

usual to consider the distance to the diagonal $\alpha_d - \alpha_b$ as a proxy for the statistical significance of the corresponding feature. The most important foundational result in the subject establishes that persistence diagrams are *stable* — they will not change much when computed on a small perturbation $\tilde{f}$ of the function $f$ (Cohen-Steiner et al., 2007).

However, an inherent limitation of the formulation of persistent homology is that it can handle only a single filtration parameter $f$. However, in practice it is common that one has to deal with multiple parameters. This translates into multiple filtration functions: a standard example is given in Figure 1, where both feature scale and density functions are necessary to obtain meaningful topological representation of a noisy point cloud. An extension of persistent homology to this more general setting is called *multiparameter* persistent homology (Botnan & Lesnick, 2022; Carlsson & Zomorodian, 2009), since roughly speaking it amounts to studying the topological variation of a continuous *multiparameter* function $f : X \to \mathbb{R}^n$ with $n \in \mathbb{N}^*$. This setting is notoriously difficult to analyze theoretically, as there is no general decomposition theorem and hence no analogue of persistence diagrams.

Still, it remains possible to define topological invariants, even in this setting. The most common one is the so-called *rank invariant*, which describes how the topological features associated to any pair of sublevel sets $\{x \in X : f(x) \leq \alpha\}$ and $\{x \in X : f(x) \leq \beta\}$ such that $\alpha \leq \beta$ (w.r.t. the partial order in $\mathbb{R}^n$), are connected. Since it is defined as an algebraic construction, and, as such, not suitable as input for subsequent data science purposes, the task of finding appropriate *representations* of this invariant, i.e., embeddings into Hilbert spaces, is critical. Hence, a number of such representations have been defined recently (Corbet et al., 2019; Coskunuzer et al., 2021; Vipond, 2020).

However, while the rank invariant is equivalent to the persistence diagram in the single-parameter case, it is known to be much less informative in the multiparameter case, even when the function admits a decomposition: many functions have different decompositions yet same rank invariants. Therefore, all aforementioned representations can encode only limited multiparameter topological information. Instead, in this work, we define representations based on *candidate decompositions* of the function, in order to create descriptors that are strictly more powerful than the rank invariant. Indeed, while there is no general decomposition theorem, there is now a whole body of work that allows to define stable, candidate decompositions (Asashiba et al., 2019; Botnan et al., 2021; Cai et al., 2020; Dey & Xin, 2021; Loiseaux et al., 2022), that are more informative than the rank invariant and that agree with the true decomposition when it exists[1]. Our representation is motivated from this series of recent contributions, and expects one of such candidate decompositions as input. Closely related to our method is the recent contribution (Carrière & Blumberg, 2020), which also proposes a representation built from a given decomposition. However, their approach, while being efficient in practice, is a heuristic with no corresponding mathematical guarantees. In particular, it is known to be unstable: similar decompositions can lead to very different representations. Our approach can be understood as a subsequent generalization of the work of (Carrière & Blumberg, 2020), with new mathematical guarantees that allow to derive, e.g., statistical rates of convergence.

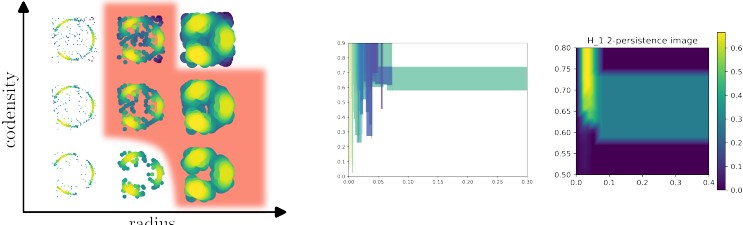

Figure 1: **(left)** Example of point cloud $X$ filtered by both feature scale (computed from unions of balls centered on $X$ with increasing radii) and (co)density. A point will be in the bifiltration if its density is high enough, and if it is close enough to $X$. Note that the circle is present in the red zone, and can be detected as a large summand in the decomposition of the corresponding module **(middle)**, or in a representation of it **(right)**.

---

[1]Strictly speaking, multiparameter persistent homology can always be decomposed in an abstract way (Botnan & Lesnick, 2022, Theorem 4.2). However, such abstract decompositions are hard to interpret and work with, hence the motivation for finding candidate decompositions that are more visual and intuitive.

**Contributions.**    Our contributions in this work are listed below:

- We provide a general framework that parametrizes representations of multiparameter persistent homology (Definition 3.1) and which encompasses previous approaches in the literature.

- We identify parameters in this framework that result in representations that have stability guarantees while still encoding more information than the rank invariant (see Theorem 3.4).

- We illustrate the performance of our framework with numerical experiments: (1) We demonstrate the practical consequences of the stability theorem by measuring the statistical convergence of our representations. (2) We achieve the best performance with the lowest runtime on several classification tasks on public data sets (see Sections 4.1 and 4.2).

**Outline.**    Our work is organized as follows. In Section 2, we recall the basics of multiparameter persistent homology. Next, in Section 3 we present our general framework and state our associated stability result. Finally, we showcase the numerical performances of our representations in Section 4, and we conclude in Section 5.

## 2    BACKGROUND

In this section, we briefly recall the basics of single and multiparameter persistent homology, and refer the reader to (Oudot, 2015; Rabadán & Blumberg, 2019) for a thorough treatment.

**Persistent homology.**    A standard procedure for studying a data set provided as a finite metric space, i.e., a set $X$ with a distance $d_X \colon X \times X \to \mathbb{R}$, is to form a graph with vertex set $X$ and an each edge connecting $x_i$ and $x_j$ when $d_X(x_i, x_j) < \epsilon$. Persistent homology is a simultaneous generalization of this procedure, replacing the graph with a *simplicial complex* (a combinatorial model of a topological space) and aggregating information across different scale parameters $\epsilon$.

More generally, given a filtered topological space $X$, by which we mean a topological space $X$ together with a function $f \colon X \to \mathbb{R}$, the system of inverse images $F(\alpha) := f^{-1}((-\infty, \alpha])$ fit together to produce a system $F(\alpha_1) \subseteq F(\alpha_2) \subseteq \ldots \subseteq F(\alpha_i) \subseteq \ldots$, where $\alpha_i \leq \alpha_{i+1}$. This system is an example of *filtration* of $X$, where a filtration is generally defined as a sequence of nested subspaces $X_1 \subseteq \ldots \subseteq X_i \subseteq \ldots \subseteq X$. Then, one can apply the $k$th *homology functor* $H_k$ on each $F(\alpha_i)$. We do not define the homology functor explicitly here, but simply recall that each $H_k(F(\alpha_i))$ is a vector space, whose basis elements represent the $k$th dimensional topological features of $F(\alpha_i)$ (connected components for $k = 0$, loops for $k = 1$, spheres for $k = 2$, etc). Moreover the inclusions $F(\alpha_i) \subseteq F(\alpha_{i+1})$ translate into linear maps $H_k(F(\alpha_i)) \to H_k(F(\alpha_{i+1}))$, which connect the features of $F(\alpha_i)$ and $F(\alpha_{i+1})$ together. Such a sequence of vector spaces connected with linear maps is a *persistence module*, which is generally defined as a functor from $(\mathbb{R}, \leq)$ (regarded as a category) to **Vec**, the category of vector spaces.

**Multiparameter persistent homology.**    The persistence modules defined above extend straightforwardly when there are multiple filtration directions. An $n$-filtration, or multifiltration, is a family of subspaces $\mathcal{F} = \{X_\alpha \subseteq X\}_{\alpha \in \mathbb{R}^n}$ such that $X_\alpha \subseteq X_{\alpha'}$ whenever $\alpha \leq \alpha'$ (in the partial order of $\mathbb{R}^n$). Moreover, we say that a multifiltration is *1-critical* if there exists a function $f \colon X \to \mathbb{R}^n$ such that $X_\alpha$ is homeomorphic to the sublevel set $\{x \in X : f(x) \leq \alpha\}$ (note that single parameter filtrations are always 1-critical). Again, applying the homology functor $H_k$ on a multifiltration $\mathcal{F}$ induces a *multiparameter persistence module* $\mathbb{M} = \mathbb{M}(\mathcal{F})$. The *support* of $\mathbb{M}$ is then defined as the set $\operatorname{supp}(\mathbb{M}) := \{\alpha \in \mathbb{R}^n : \dim(\mathbb{M}(\alpha)) > 0\}$. Any module which is of dimension 1 everywhere on its support is called an *indicator module*, and, in addition, when its support is an interval of $\mathbb{R}^n$ (Botnan & Lesnick, 2022, Definition 2.1), it is called an *interval module*.

A multiparameter persistence module $\mathbb{M}$ can always be decomposed in a unique way: $\mathbb{M} = \oplus_{i=1}^m M_i$ (Botnan & Lesnick, 2022, Theorem 4.2), where the $M_i$ are indecomposable modules that are called the *summands* of $\mathbb{M}$. See Figure 1 (middle) for an example of decomposition. For single parameter filtrations, the summands of the decomposition are known to be interval modules $\mathbb{M} = \oplus_{i=1}^m \mathbb{I}[\alpha_{b_i}, \alpha_{d_i}]$, and can be summarized in a *persistence barcode* $D(\mathbb{M}) = \{(\alpha_{b_i}, \alpha_{d_i})\}_{1 \leq i \leq m}$. This result does not extend to the multiparameter setting however since summands can have dimension

larger than 1 on their support. Hence, the *rank invariant* has been introduced as a weaker invariant for such modules: it is defined as the function $\mathrm{RI} : (\alpha, \beta) \mapsto \mathrm{rank}(\mathbb{M}(\alpha) \to \mathbb{M}(\beta))$ for any $\alpha \leq \beta$.

Finally, multiparameter persistence modules can be compared with two standard distances: the *interleaving* and *bottleneck* (or $\ell^\infty$) distances. Their explicit definitions are technical and not necessary for our main exposition, so we refer the reader to, e.g., (Botnan & Lesnick, 2022, Sections 6.1, 6.4) for more details. The *stability* theorem (Lesnick, 2015, Theorem 5.3) states that 1-critical multiparameter persistence modules are stable: $d_I(\mathbb{M}, \mathbb{M}') \leq \|f - f'\|_\infty$, where $f$ and $f'$ are functions associated to the 1-critical modules $\mathbb{M}$ and $\mathbb{M}'$ respectively.

# 3 A FRAMEWORK FOR PERSISTENCE MODULE REPRESENTATION

Even though multiparameter persistence modules are known to encode useful data information, their algebraic definitions make them not suitable for subsequent data science and machine learning purposes. Hence, in this section, we propose a general framework for studying *representations* of multiparameter persistence modules, i.e., maps defined on the space of multiparameter persistence modules and taking values in an (implicit or explicit) Hilbert space. Defining such maps is crucial for subsequent data science tasks, as most of the standard methods in statistical machine learning expect the input to be Euclidean vectors, or at least to be equipped with a scalar product.

## 3.1 PERSISTENCE MODULE REPRESENTATION

**Framework.** In this article, we study the following general representation framework:

**Definition 3.1.** Let $\mathbb{M}$ be a multiparameter persistence module. Let $\mathbb{M} = \oplus_{i=1}^m M_i$ be its decomposition (note that the $M_i$ need not be intervals), and let $\mathcal{M}$ be the space of indecomposable persistence modules. A *multiparameter representation* of $\mathbb{M}$ is:

$$V_{\mathrm{op}, w, \phi}(\mathbb{M}) = \mathrm{op}(\{w(M_i) \cdot \phi(M_i)\}_{i=1}^m), \tag{1}$$

where op is a permutation invariant operation (sum, max, min, mean, etc), $w : \mathcal{M} \to \mathbb{R}$ is a weight function, and $\phi : \mathcal{M} \to \mathcal{H}$ sends any indecomposable module to a vector in a Hilbert space $\mathcal{H}$.

This general definition is inspired from a similar framework that was introduced for single-parameter persistence with the automatic representation method *Perslay* (Carrière et al., 2020).

*Remark* 3.2. Definition 3.1 requires being able to decompose multiparameter persistence modules. Several approaches have been proposed for this problem in the literature, that provide either exact (Dey & Xin, 2022; Lesnick & Wright, 2015) or approximate (Loiseaux et al., 2022) decompositions, or decompositions that are restricted to specific homology dimensions (Cai et al., 2020).

**Relation to previous work.** Interestingly, specific choices of op, $w$ and $\phi$ can reproduce the previous work from the literature. For sake of simplicity, assume $\mathbb{M} = \oplus_{i=1}^m M_i$ can be decomposed into indicator modules (we can thus identify any summand $M_i$ with its support), then:

- Using $w : M_i \mapsto 1$, $\phi : M_i \mapsto \left\{ \begin{array}{ll} \mathbb{R}^n & \to \mathbb{R} \\ x & \mapsto \Lambda(x, M_i\big|_{\ell_x}) \end{array} \right.$ and op $= k$th maximum, where $l_x$ is the diagonal line crossing $x$, and $\Lambda(\cdot, \ell)$ denotes the tent function associated to any segment $\ell \subset \mathbb{R}^n$, induces the $k$th multiparameter persistence landscape (Vipond, 2020).

- Using $w : M_i \mapsto 1$, $\phi : M_i \mapsto \left\{ \begin{array}{ll} \mathbb{R}^n \times \mathbb{R}^n & \to \mathbb{R}^d \\ p, q & \mapsto w'(M_i \cap [p, q]) \cdot \phi'(M_i \cap [p, q]) \end{array} \right.$ and op $= \mathrm{op}'$, where $\mathrm{op}'$, $w'$ and $\phi'$ are the parameters of any persistence diagram representation from Perslay, induces the multiparameter persistence kernel (Corbet et al., 2019).

- Using $w : M_i \mapsto \mathrm{vol}(M_i)$, $\phi : M_i \mapsto \left\{ \begin{array}{ll} \mathbb{R}^n & \to \mathbb{R} \\ x & \mapsto \exp(-\min_{\ell \in L} d(x, M_i\big|_\ell)^2/\sigma^2) \end{array} \right.$ and op $= \sum$, where $L$ is a set of (pre-defined) diagonal lines, induces the multiparameter persistence image (Carrière & Blumberg, 2020).

Concerning the first two approaches, it is worth noting that they do not depend on the decompositions and the global geometric shapes of the support of the summands since they are equivalent to the rank

invariant. In particular, it is easy to build different modules with same rank invariants and thus same landscapes and kernels (see (Loiseaux et al., 2022, Section 7.1) for examples). On the other hand, the third approach does use the module decomposition but is known to be unstable (Figure 11 in (Botnan & Lesnick, 2018) provides two modules that are close in the interleaving distance but that also have multiparameter persistence images that are far away from each other). In the next section, we study representations that allow to encode the summand shapes while also enjoying stability guarantees.

## 3.2 METRIC PROPERTIES

In this section, we study specific parameters within our framework in Definition 3.1, and prove associated theoretical properties about their associated representations.

**Definition.** We begin by defining parameters that we call *geometric and stable* (GS) in the sense that they capture shape information about the summands, and that they also lead to robust representations.

**Definition 3.3.** The *geometric and stable* (GS) representation parameters are:

1. the weight function $w : M \mapsto \sup\{\varepsilon > 0 : \exists y \in \mathbb{R}^n \text{ s.t. } \ell_{y,\varepsilon} \subset \text{supp}\,(M)\}$, where $\ell_{y,\varepsilon}$ is the segment between $y - \varepsilon \cdot [1, \ldots, 1]$ and $y + \varepsilon \cdot [1, \ldots, 1]$,

2. the individual summand representations $\phi_\delta(M) : \mathbb{R}^n \to \mathbb{R}$:

   (a) $\phi_\delta(M)(x) = \frac{1}{\delta} w(\text{supp}\,(M) \cap R_{x,\boldsymbol{\delta}})$,   (b) $\phi_\delta(M)(x) = \frac{1}{(2\delta)^n} \text{vol}\,(\text{supp}\,(M) \cap R_{x,\boldsymbol{\delta}})$,

   (c) $\phi_\delta(M)(x) = \frac{1}{(2\delta)^n} \sup_{x',\boldsymbol{\delta}'} \{\text{vol}(R_{x',\boldsymbol{\delta}'}) : R_{x',\boldsymbol{\delta}'} \subseteq \text{supp}\,(M) \cap R_{x,\boldsymbol{\delta}}\}$,

   where $R_{x,\boldsymbol{\delta}}$ is the hypersquare $\{y \in \mathbb{R}^n : x - \boldsymbol{\delta} \leq y \leq x + \boldsymbol{\delta}\} \subseteq \mathbb{R}^n$, $\boldsymbol{\delta} := \delta \cdot [1, \ldots, 1] \in \mathbb{R}^n$ for any $\delta > 0$, and vol denotes the volume of a set in $\mathbb{R}^n$.

3. the permutation invariant operators $\text{op} = \sum$ and $\text{op} = \sup$.

In other words, our weight function is the length of the largest diagonal segment one can fit inside $\text{supp}\,(M)$, and summand representations (a), (b) and (c) are largest diagonal length, volume, and largest hypersquare volume one can fit locally inside $\text{supp}\,(M) \cap R_{x,\boldsymbol{\delta}}$ respectively.

Equipped with these GS parameters, we can now define the two following GS representations, that can be applied on any multiparameter persistence module $\mathbb{M} = \oplus_{i=1}^m M_i$:

$$V_{p,\delta}(\mathbb{M}) := \sum_{i=1}^m \frac{w(M_i)^p}{\sum_{j=1}^m w(M_j)^p} \phi_\delta(M_i), \text{ for some } p \in \mathbb{R}_+, \tag{2}$$

$$V_{\infty,\delta}(\mathbb{M}) := \sup_{1 \leq i \leq m} \phi_\delta(M_i). \tag{3}$$

These GS parameters allow for some trade-off between computational cost and the amount of information that is kept: (a) and (c) are very easy to compute on interval modules, but (b) encodes more information about summand shapes. See Figures 1 (right) and 2b for visualizations.

**Stability.** The main motivation for introducing our GS representations is that one can show that they are stable in the interleaving and bottleneck distances, as stated in the following theorem. We assume that persistence modules have support included in a compact set $K \subseteq \mathbb{R}^n$ for simplicity (but our results extend straightforwardly to modules restricted to $K$ if their supports are infinite).

**Theorem 3.4.** *Let $\mathbb{M} = \oplus_{i=1}^m M_i$ and $\mathbb{M}' = \oplus_{j=1}^{m'} M_j'$ be two multiparameter persistence modules that can be decomposed with indicator modules. Assume that we have $\frac{1}{m} \sum_i w(M_i), \frac{1}{m'} \sum_j w(M_j') \geq C$, for some $C > 0$. Then for any parameter $\delta > 0$, one has*

$$\|V_{0,\delta}(\mathbb{M}) - V_{0,\delta}(\mathbb{M}')\|_\infty \leq 2(d_\text{B}(\mathbb{M}, \mathbb{M}') \wedge \delta)/\delta, \tag{4}$$

$$\|V_{1,\delta}(\mathbb{M}) - V_{1,\delta}(\mathbb{M}')\|_\infty \leq \left[4 + \frac{2}{C}\right](d_\text{B}(\mathbb{M}, \mathbb{M}') \wedge \delta)/\delta, \tag{5}$$

$$\|V_{\infty,\delta}(\mathbb{M}) - V_{\infty,\delta}(\mathbb{M}')\|_\infty \leq (d_\text{I}(\mathbb{M}, \mathbb{M}') \wedge \delta)/\delta, \tag{6}$$

These results are the main theoretical contribution in this work, as the only other decomposition-based representation in the literature (Carrière & Blumberg, 2020) has no such guarantees. The other representations (Corbet et al., 2019; Coskunuzer et al., 2021; Vipond, 2020) enjoy similar guarantees than ours, but are computed from the rank invariant and do not exploit the information contained in decompositions. Theorem 3.4 shows that our GS representations bring the best of both worlds: they are richer than the rank invariant and stable at the same time.

A proof of Theorem 3.4 can be found in Appendix C. One might worry that the minimum with $\delta$ encountered in the upper bounds in Theorem 3.4 show that our GS representations are too stable and not discriminative enough for practical purposes; while we did not observe such behaviour in our numerical experiments, we also provide an additional stability result with a similar, yet slightly different representation in Appendix B, with an associated upper bound without minimum.

*Remark* 3.5. Our GS representations are injective on indicator modules: if the support of two modules are different, then the GS representations (evaluated on a point that belongs to the support of one summand but not on the support of the other) will differ, provided that $\delta$ is sufficiently small.

## 4 NUMERICAL EXPERIMENTS

In this section, we illustrate our GS representations with numerical experiments. First, we explore the stability theorem in Section 4.1 by studying the convergence rates of our GS representations on various data sets. Then, we showcase the efficiency of our GS representations on classification tasks in Section 4.2. Our code for computing GS representations is based on the MMA (Loiseaux et al., 2022) and Gudhi (The GUDHI Project, 2022) libraries and is publicly available[2], and described in Appendix D. We also provide a few (favorable) running time comparisons in Appendix E.

**Decomposition.** Our GS representations require to compute a persistence module decomposition first. Many different approaches can be used for this task (Loiseaux et al., 2022; Dey & Xin, 2022; Cai et al., 2020; Lesnick & Wright, 2015). In our experiments, we always use the MMA approximate decomposition (Loiseaux et al., 2022) because of its simplicity and rapidity.

### 4.1 CONVERGENCE RATE

In this section, we study the convergence rate of our representation with respect to the number of sampled points. Similar to the single parameter persistence setting (Chazal et al., 2015), these rates are derived for free from our stability theorem. Indeed, since concentration inequalities for multiparameter persistence modules have already been showed in the literature, these concentration inequalities apply directly to our representations. Note that while Equations (7) and (8), which provide such rates, are stated for GS representation (3), they also hold for GS representation (2).

**Measure bifiltration.** Let $\mu$ be a compactly supported probability measure of $\mathbb{R}^D$, and let $\mu_n$ be the discrete measure associated to a sampling of $n$ points from $\mu$. The *measure bifiltration* associated to $\mu$ and $\mu_n$ is defined as $\mathcal{F}_{r,t}^\mu := \{x \in \mathbb{R}^D : \mu(B(x,r)) \leq t\}$, where $B(x,r)$ denotes the Euclidean ball centered on $x$ with radius $r$. Now, let $\mathbb{M}$ and $\mathbb{M}_n$ be the multiparameter persistence modules obtained from applying the homology functor on top of the measure bifiltrations $\mathcal{F}^\mu$ and $\mathcal{F}^{\mu_n}$. These modules are known to enjoy the following stability result (Blumberg & Lesnick, 2021, Theorem 3.1, Proposition 2.23 (i)):

$$d_{\mathrm{I}}(\mathbb{M}, \mathbb{M}_n) \leq d_{\mathrm{Pr}}(\mu, \mu_n) \leq \min(d_W^p(\mu, \mu_n)^{\frac{1}{2}}, d_W^p(\mu, \mu_n)^{\frac{p}{p+1}}),$$

where $d_W^p$ and $d_{\mathrm{Pr}}$ stand for the $p$-Wasserstein and Prokhorov distances between probability measures. Combining these inequalities with Theorem 3.4, then taking expectations and applying the concentration inequalities of the Wasserstein distance (see (Lei, 2020, Theorem 3.1) and (Fournier & Guillin, 2015, Theorem 1)) lead to:

$$\delta \mathbb{E}\left[\|V_{\infty,\delta}(\mathbb{M}) - V_{\infty,\delta}(\mathbb{M}_n)\|_\infty\right] \leq \left(c_{p,q}\mathbb{E}\left(|X|^q\right) n^{-\left(\frac{1}{2p \vee d}\right) \wedge \frac{1}{p} - \frac{1}{q}} \log^{\alpha/q} n\right)^{\frac{p}{p+1}}, \quad (7)$$

where $\alpha = 2$ if $2p = q = d$, $\alpha = 1$ if $d \neq 2p$ and $q = dp/(d-p) \wedge 2p$ or $q > d = 2p$ and $\alpha = 0$ otherwise, $c_{p,q}$ is a constant that depends on $p$ and $q$, and $X$ is a random variable of law $\mu$.

---

[2] `hidden_for_double_blind_reviews`

**Čech complex and density.** A limitation of the measure bifiltration is that it can be difficult to compute. Hence, we now focus on another, easier to compute bifiltration. Let $X$ be a compact $d$-submanifold of $\mathbb{R}^D$ ($d \leq D$), and $\mu$ be a measure on $X$ with density $f$ with respect to the uniform measure on $X$. We now define the 1-critical bifiltration $\mathcal{F}^{C,f}$ with:

$$\mathcal{F}^{C,f}_{u,v} := \text{Čech}(u) \cap f^{-1}([v,\infty)) = \left\{ x \in \mathbb{R}^D : d(x,X) \leq u, f(x) \geq v \right\}.$$

Moreover, given a set $X_n$ of $n$ points sampled from $\mu$, we also consider the approximate bifiltration $\mathcal{F}^{C,f_n}$, where $f_n \colon X \to \mathbb{R}$ is an estimation of $f$ (such as, e.g., a kernel density estimator). Let $\mathbb{M}$ and $\mathbb{M}_n$ be the multiparameter persistence modules associated to $\mathcal{F}^{C,f}$ and $\mathcal{F}^{C,f_n}$. Then, the stability of the interleaving distance (Lesnick, 2015, Theorem 5.3) ensures:

$$d_I(\mathbb{M}, \mathbb{M}_n) \leq \|f - f_n\|_\infty \vee d_H(X, X_n),$$

where $d_H$ stands for the Hausdorff distance. Moreover, concentration inequalities for the Hausdorff distance and kernel density estimators are also available in the literature under some conditions (see (Chazal et al., 2015, Theorem 4) and (Kim et al., 2019, Corollary 15)). More precisely, when the density $f$ is $L$-Lipschitz and bounded from above and from below, i.e., when $0 < f_{\min} \leq f \leq f_{\max} < \infty$, and when $f_n$ is a kernel density estimator of $f$ with associated kernel $k$, one has:

$$\mathbb{E}(d_H(X, X_n)) \lesssim \left( \frac{\log n}{n} \right)^{\frac{1}{d}} \text{ and } \mathbb{E}(\|f - f_n\|_\infty) \lesssim Lh_n \mathbb{E}(\|K\|_\infty) + \sqrt{\frac{\log(1/h_n)}{nh_n^d}},$$

where $h_n$ is the (adaptive) bandwidth of the kernel $k$, and $K$ is a random variable of density $k$. In particular, if $\mu$ is a measure comparable to the uniform measure of a $d = 2$-manifold, then for any stationary sequence $h_n := h > 0$, and considering a Gaussian kernel $k$, one has:

$$\delta \mathbb{E}\left[ \|V_{\infty,\delta}(\mathbb{M}) - V_{\infty,\delta}(\mathbb{M}_n)\|_\infty \right] \lesssim \sqrt{\frac{\log n}{n}} + L\sqrt{2\log(d)}h. \tag{8}$$

**Empirical convergence rates.** Now that we have established the theoretical convergence rates of our GS representations, we estimate and validate them on data sets. We will study a synthetic data set and a real data set of point clouds obtained with immunohistochemistry.

*Annulus with non-uniform density.* In this synthetic example, we generate an annulus in $\mathbb{R}^2$ with a non-uniform density, displayed in Figure 2a. Then, we compute the bifiltration $\mathcal{F}^{C,f_n}$ corresponding to the Alpha filtration and the sublevel set filtration of a kernel density estimator, with bandwidth parameter $h = 0.1$, on the complete Alpha simplicial complex. Finally, we compute the representations (see Figure 2b) of the associated multiparameter module (in homology dimension 1), and their normalized distances to the target representation, using either $\|\cdot\|_2^2$ or $\|\cdot\|_\infty$. The corresponding distances for various number of sample points are displayed in log-log plots in Figure 2c. One can see that the empirical rate is roughly consistent with the theoretical one ($-1/2$ for $\|\cdot\|_\infty$ and $-1$ for $\|\cdot\|_2$), even when $p \neq \infty$ (in which case our GS representations are stable for $d_B$ but not for $d_I$).

*Immunohistochemistry data.* In our second experiment, we consider a point cloud representing cells, taken from (Vipond et al., 2021), see Figure 3a. These cells are given with biological markers, which are typically used to assess, e.g., cell types and functions. In this experiment, we first triangulate the point cloud by placing a $100 \times 100$ grid on top of it. Then, we filter this grid using the sublevel set filtrations of kernel density estimators (with Gaussian kernel and bandwidth $h = 1$) associated to the CD8 and CD68 biological markers for immune cells. Finally, we compute the associated multiparameter modules in homology dimensions 0 and 1, and we compute and concatenate their corresponding GS representations. The convergence rates are displayed in Figure 3b. Similar to the previous experiment, the theoretical convergence rate of our representations is upper bounded by the one for kernel density estimators with the $\infty$-norm, which is of order $\frac{1}{\sqrt{n}}$ with respect to the number $n$ of sampled points. Again, the observed convergence rate are consistent with the theoretical ones.

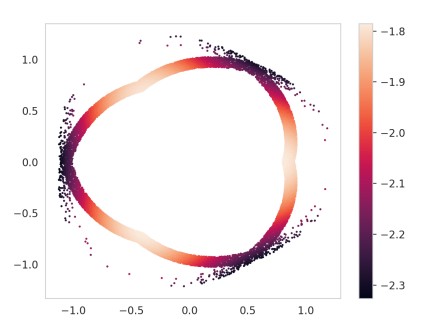
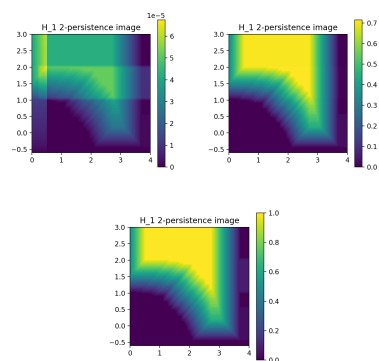

(a) Scatter plot of the synthetic data set, comprised of 25,000 points, and colored by a kernel density estimator with bandwidth $h = 0.5$.

(b) GS representations with parameters, $p = 0$ (upper left), $p = 1$ (upper right), $p = \infty$ (bottom).

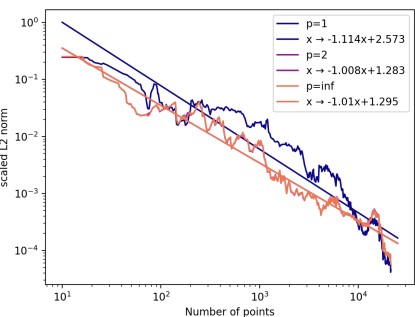
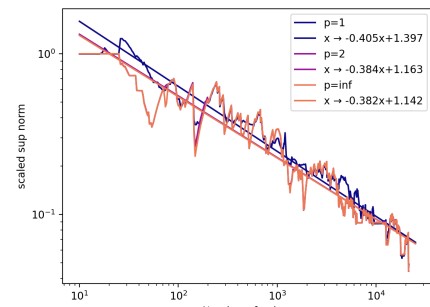

(c) **(left)** $\|\cdot\|_2^2$ and **(right)** $\|\cdot\|_\infty$ (right) between the target representation and the empirical one w.r.t. $n$.

Figure 2: Convergence rate of synthetic data set.

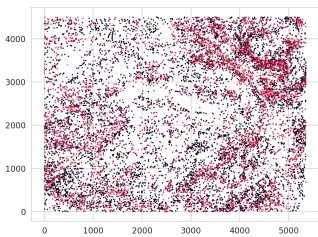
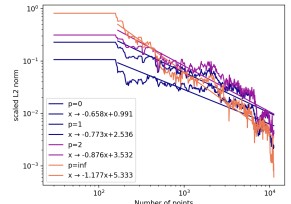
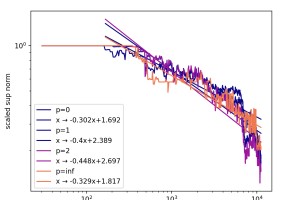

(a) Point cloud of cells colored by CD8 (red) and CD68 (black).

(b) **(left)** $\|\cdot\|_2^2$ and **(right)** $\|\cdot\|_\infty$ distances between the target representation and the empirical one w.r.t. $n$.

Figure 3: Convergence rate of immunofluorescence data set.

## 4.2 CLASSIFICATION

We illustrate the efficiency of our GS representations by using them for classification purposes. We show that they perform comparably or better than existing topological representations as well as standard baselines on UCR benchmark data sets. We work with point clouds obtained from time delay embedding on time series from the UCR data sets, following the procedure of (Carrière & Blumberg, 2020). Furthermore, we show that on the immunohistochemistry data set of point clouds from the previous section, where each point cloud represents single cells of a given type (CD8, CD68, etc), our method is significantly better than the competition. In both tasks, every point cloud has a label (corresponding, e.g., to the type of its cells in the immunohistochemistry data), and our goal is to check whether we can predict these labels using the bifiltration defined in the previous section.

We also compare the performances of our GS representations (evaluated on a $50 \times 50$ grid) to the one of the multiparameter persistence landscape (MPL) (Vipond, 2020), kernel (MPK) (Corbet et al., 2019) and images (MPI) (Carrière & Blumberg, 2020) [3]. We also compare to some non-topological baselines: we used the standard Ripley function evaluated on 100 evenly spaced samples in $[0, 1]$ for immunohistochemistry data, and k-NN classifiers with three difference distances for the UCR time series (denoted by B1, B2, B3), as suggested in (Dau et al., 2018). All scores on the immunohistochemistry data were computed with 5 folds after cross-validating a few classifiers (random forests, support vector machines and xgboost). We also evaluated the performances of a few different GS representation parameters $p$ and $\delta$ separately. For the time series data, our accuracy scores were obtained after also cross-validating the following GS parameters; $p \in \{0, 1\}$, op $\in \{\mathrm{sum}, \mathrm{mean}\}$, $\delta \in \{0.01, 0.1, 0.5, 1\}$, $h \in \{0.1, 0.5, 1, 1.5\}$ with homology dimensions 0 and 1. All results can be found in Tables 1 (immunohistochemistry) and 2 (time series—note that there are no variances since train/test splits were provided in the data sets). One can see that our GS representations almost always outperform topological baselines, and are comparable to the standard baselines on the UCR benchmarks. Most notably, all tested GS parameters showed improvement over the baselines for the immunohistochemistry data.

| Ours | $p$ | $\delta$ | Accuracy (%) |
|---|---|---|---|
| | | $10^{-2}$ | $74.1 \pm 2.9$ |
| - | 1 | $10^{-3}$ | $75.6 \pm 3.4$ |
| | | $10^{-4}$ | $68.7 \pm 3.0$ |
| | | $10^{-2}$ | **77.81± 2.0** |
| - | 0 | $10^{-3}$ | $76.0 \pm 3.3$ |
| | | $10^{-4}$ | $70.5 \pm 3.0$ |
| MPL | - | - | $60.5 \pm 3.5$ |
| Ripley | - | - | $67.2 \pm 2.3$ |

Table 1: Accuracy scores for immunohisto-chemistry data.

| Dataset | B1 | B2 | B3 | MPK | MPL | MPI | Ours |
|---|---|---|---|---|---|---|---|
| DPOAG | 62.6 | 62.6 | **77.0** | 67.6 | 70.5 | **71.9** | **71.9** |
| DPOC | 71.7 | **72.5** | 71.7 | **74.6** | 69.6 | 71.7 | 73.8 |
| DPTW | **63.3** | **63.3** | 59.0 | 61.2 | 56.1 | 61.9 | 62.6 |
| PPOAG | 78.5 | 78.5 | **80.5** | 78.0 | 78.5 | 81.0 | **81.9** |
| PPOC | **80.8** | 79.0 | 78.4 | 78.7 | 78.7 | **81.8** | 79.4 |
| PPTW | 70.7 | **75.6** | **75.6** | **79.5** | 73.2 | 76.1 | 75.6 |
| ECG200 | **88.0** | **88.0** | 77.0 | 77.0 | 74.0 | **83.0** | **83.0** |
| IPD | **95.5** | **95.5** | 95.0 | 80.7 | 78.6 | 71.9 | **81.2** |
| MI | 68.4 | **74.7** | 73.7 | 55.4 | 55.7 | 60.0 | **64.3** |
| P | 96.2 | **100.0** | **100.0** | 92.4 | 84.8 | 97.1 | **99.0** |
| SL | 78.9 | **84.6** | 79.2 | 78.2 | 64.6 | **83.8** | **83.8** |
| GP | **91.3** | **91.3** | 90.7 | 88.7 | 94.0 | 90.7 | **96.3** |
| GPAS | 89.9 | **96.5** | 91.8 | **93.0** | 85.1 | 90.5 | 88.0 |
| GPMVF | 97.5 | 97.5 | **99.7** | **96.8** | 88.3 | 95.9 | 95.3 |
| GPOVY | 95.2 | **96.5** | 83.8 | 99.0 | 97.1 | **100.0** | **100.0** |
| PC | **93.3** | 92.2 | 87.8 | 85.6 | 84.4 | 86.7 | **93.1** |
| SC | 88.0 | 98.3 | **99.3** | 50.7 | 60.3 | 60.0 | **61.3** |

Table 2: Accuracy scores for time series.

## 5 CONCLUSION

In this article, we study the general question of representing multiparameter persistence modules in Topological Data Analysis. We first introduce a general theoretical framework that allows to recover existing approaches in the literature, as well as to define new ones. Then, we identify and prove the stability of some specific representations in that framework, that we call *geometric and stable* (GS) representations since they are able to encode the shapes of the summands comprised in the modules, making these representations more powerful than already existing ones. We finally showcase the stability and efficiency of these GS representations in various theoretical and numerical applications about statistical convergence and classification accuracies.

Several questions remain open for future works:

- The framework that we introduced in Definition 3.1 is similar to the Perslay framework of single parameter persistence (Carrière et al., 2020). In this work, each of the framework parameter was optimized by a neural network. It is thus natural to investigate whether one can optimize the module representation within our framework in a data-driven way as well.

- In our numerical applications, we focused on representations computed off of MMA decompositions (Loiseaux et al., 2022). In the future, we plan to investigate whether working with other decomposition methods (Cai et al., 2020; Dey & Xin, 2022) lead to better numerical performances when combined with our representation framework.

- Even though our experiments were done on data sets equipped with bifiltrations, our code and framework is well-defined for any number of filtrations. Investivating the dependences and performances of our framework on more than two filtrations is thus of particular interest.

---

[3]Note that the sizes of the point clouds obtained from immunohistochemistry were too large for MPK and MPI using the code provided in `https://github.com/MathieuCarriere/multipers`.

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

## A    ADDITIONAL BACKGROUND

**Definition A.1.** Let $n \in \mathbb{N}^*$ and $\mathcal{F} = \{\mathcal{F}^{(1)}, \ldots, \mathcal{F}^{(n)}\}$ be a multi-filtration on a topological space $X$. Let $\mathbf{e}, b \in \mathbb{R}^n$, and $\ell_{\mathbf{e},b} : \mathbb{R} \to \mathbb{R}^n$ be the line in $\mathbb{R}^n$ defined with $\ell_{\mathbf{e},b}(t) = t \cdot \mathbf{e} + b$, that is, $\ell_{\mathbf{e},b}$ is the line of direction $\mathbf{e}$ passing through $b$. Let $\mathcal{F}_{\mathbf{e},b} : \mathbb{R} \to \mathcal{P}(X)$ defined with $\mathcal{F}_{\mathbf{e},b}(t) = \bigcap_{i=1}^{n} \mathcal{F}^{(i)}([\ell_{\mathbf{e},b}(t)]_i)$, where $[\cdot]_i$ denotes the $i$-th coordinate. Then, each $\mathcal{F}_{\mathbf{e},b}$ is a single-parameter filtration and has a corresponding persistence diagram, or barcode $B_{\mathbf{e},b}$. The set $\mathcal{B}(\mathcal{F}) = \{B_{\mathbf{e},b} : \mathbf{e}, b \in \mathbb{R}^n\}$ is called the *fibered barcode* of $\mathcal{F}$.

The two following lemmas from (Landi, 2018) describe two useful properties of the fibered barcode.

**Lemma A.2** (Lemma 1 in (Landi, 2018)). *Let $\mathbf{e}, b \in \mathbb{R}^n$ and $\ell_{\mathbf{e},b}$ be the corresponding line. Let $\hat{e} = \min_i[\mathbf{e}]_i$. Let $\mathcal{F}, \mathcal{F}'$ be two multi-filtrations, $\mathbb{M}, \mathbb{M}'$ be the corresponding persistence modules and $B_{\mathbf{e},b} \in \mathcal{B}(\mathcal{F})$ and $B'_{\mathbf{e},b} \in \mathcal{B}(\mathcal{F}')$ be the corresponding barcodes in the fibered barcodes of $\mathcal{F}$ and $\mathcal{F}'$. Then, the following stability property holds:*

$$d_{\mathrm{B}}(B_{\mathbf{e},b}, B'_{\mathbf{e},b}) \leq \frac{d_{\mathrm{I}}(\mathbb{M}, \mathbb{M}')}{\hat{e}}. \tag{9}$$

**Lemma A.3** (Lemma 2 in (Landi, 2018)). *Let $\mathbf{e}, \mathbf{e}', b, b' \in \mathbb{R}^n$ and $\ell_{\mathbf{e},b}, \ell_{\mathbf{e}',b'}$ be the corresponding lines. Let $\hat{e} = \min_i[\mathbf{e}]_i$ and $\hat{e}' = \min_i[\mathbf{e}']_i$. Let $\mathcal{F}$ be a multi-filtration, $\mathbb{M}$ be the corresponding persistence module and $B_{\mathbf{e},b}, B_{\mathbf{e}',b'} \in \mathcal{B}(\mathcal{F})$ be the corresponding barcodes in the fibered barcode of $\mathcal{F}$. Assume $\mathbb{M}$ is decomposable $\mathbb{M} = \oplus_{i=1}^{m} M_i$, and let $K > 0$ such that $M_i \subseteq B_{\infty}(0, K) := \{x \in \mathbb{R}^n : \|x\|_{\infty} \leq K\}$ for all $i \in [\![1, m]\!]$. Then, the following stability property holds:*

$$d_{\mathrm{B}}(B_{\mathbf{e},b}, B_{\mathbf{e}',b'}) \leq \frac{(K + \max\{\|b\|_{\infty}, \|b'\|_{\infty}\}) \cdot \|\mathbf{e} - \mathbf{e}'\|_{\infty} + \|b - b'\|_{\infty}}{\hat{e} \cdot \hat{e}'}. \tag{10}$$

## B    AN ADDITIONAL STABILITY THEOREM

In this section, we define a new GS representation, with a slightly different type of upper bound. It relies on the notations introduced in Appendix A. We also slightly abuse notations and use $M_i$ to denote both an indicator module and its support.

**Proposition B.1.** *Let $\sigma > 0$, and let $0 \leq \delta \leq \delta(\mathbb{M})$, where*

$$\delta(\mathbb{M}) := \inf\{\delta \geq 0 : \Gamma_{\mathbb{M}} \text{ achieves } d_{\mathrm{B}}(B_{\mathbf{e}_\Delta, x}, B_{\mathbf{e}_\Delta, x+\delta\mathbf{u}}) \text{ for all } x, \mathbf{u} \text{ s.t. } \|\mathbf{u}\|_{\infty} = 1, \langle \mathbf{e}_\Delta, \mathbf{u} \rangle = 0\},$$

*where $\Gamma_{\mathbb{M}}$ is the partial matching induced by the decomposition of $\mathbb{M}$. Let $\mathcal{N}(x, \sigma)$ denote the function $\mathcal{N}(x, \sigma) : \begin{cases} \mathbb{R}^n & \to \mathbb{R} \\ p & \mapsto \exp\left(-\frac{\|p-x\|^2}{2\sigma^2}\right) \end{cases}$ and let*

$$V_{\delta,\sigma}(\mathbb{M}) : \begin{cases} \mathbb{R}^n & \to \mathbb{R} \\ x & \mapsto \max_{1 \leq i \leq m} \ \max_{f \in \mathcal{C}(x, \delta, M_i)} \ \|\mathcal{N}(x, \sigma) \cdot f\|_1 \end{cases} \tag{11}$$

*where $\mathcal{C}(x, \delta, M_i)$ stands for the set of indicator functions from $\mathbb{R}^n$ to $\{0, 1\}$ whose support is $T_\delta(\ell) \cap M_i$, where $\ell$ is a connecting component of $\mathrm{im}(\ell_{\mathbf{e}_\Delta, x}) \cap M_i$ and $\mathbf{e}_\Delta = [1, \ldots, 1] \in \mathbb{R}^n$, and where $T_\delta(\ell)$ is the $\delta$-thickening of the line $L(\ell)$ induced by $\ell$: $T_\delta(\ell) = \{x \in \mathbb{R}^k : \|x, L(\ell)\|_{\infty} \leq \delta\}$.*

*Then, $V_{\delta,\sigma}$ satisfies the following stability property:*

$$\|V_{\delta,\sigma}(\mathbb{M}) - V_{\delta,\sigma}(\mathbb{M}')\|_\infty \leq (\sqrt{\pi}\sigma)^n \cdot \sqrt{2^{n+1}\delta^{n-1}d_{\mathrm{I}}(\mathbb{M}, \mathbb{M}') + C_n(\delta)}, \tag{12}$$

*where $C_n(\cdot)$ is a continuous function such that $C_n(\delta) \to 0$ when $\delta \to 0$.*

*Proof.* Let $\mathbb{M} = \oplus_{i=1}^m M_i$ and $\mathbb{M}' = \oplus_{j=1}^{m'} M_j'$ be two persistence modules that are decomposable into indicators, let $x \in \mathbb{R}^k$ and let $0 \leq \delta \leq \min\{\delta(\mathbb{M}), \delta(\mathbb{M}')\}$.

**Notations.** We first introduce some notations. Let $N$ (resp. $N'$) be the number of bars in $B_{\mathbf{e}_\Delta, x}$ (resp. $B'_{\mathbf{e}_\Delta, x}$), and assume without loss of generality that $N \leq N'$. Let $\Gamma$ be the partial matching achieving $d_{\mathrm{B}}(B_{\mathbf{e}_\Delta, x}, B'_{\mathbf{e}_\Delta, x})$. Let $N_1$ (resp. $N_2$) be the number of bars in $B_{\mathbf{e}_\Delta, x}$ that are matched (resp. not matched) to a bar in $B'_{\mathbf{e}_\Delta, x}$ under $\Gamma$, so that $N = N_1 + N_2$. Finally, note that $B_{\mathbf{e}_\Delta, x} = \{\ell : \exists i \text{ such that } \ell \in \mathcal{C}(\mathrm{im}(\ell_{\mathbf{e}_\Delta, x}) \cap M_i) \text{ and } \mathrm{im}(\ell_{\mathbf{e}_\Delta, x}) \cap M_i \neq \emptyset\}$, where $\mathcal{C}$ stands for the set of connected components (and similarly for $B'_{\mathbf{e}_\Delta, x}$), and let $F_\Gamma : B_{\mathbf{e}_\Delta, x} \to B'_{\mathbf{e}_\Delta, x}$ be a function defined on all bars of $B_{\mathbf{e}_\Delta, x}$ that coincides with $\Gamma$ on the $N_1$ bars of $B_{\mathbf{e}_\Delta, x}$ that have an associated bar in $B'_{\mathbf{e}_\Delta, x}$, and that maps the $N_2$ remaining bars of $B_{\mathbf{e}_\Delta, x}$ to some arbitrary bars in the $(N' - N_1)$ remaining bars of $B'_{\mathbf{e}_\Delta, x}$.

**A reformulation of the problem with vectors.** We now derive vectors that allow to reformulate the problem in a useful way. Let $\hat{V}$ be the sorted vector of dimension $N$ containing all weights $\|\mathcal{N}(x, \sigma) \cdot f\|_1$, where $f$ is the indicator function whose support is $T_\delta(\ell) \cap M_i$ for some $M_i$, where $\ell \in B_{\mathbf{e}_\Delta, x}$ is a connected component of $\mathrm{im}(\ell_{\mathbf{e}_\Delta, x}) \cap M_i$. Now, let $\hat{V}'$ be the vector of dimension $N'$ obtained by concatenating the two following vectors:

- the vector $\hat{V}_1'$ of dimension $N$ whose $i$th coordinate is $\|\mathcal{N}(x, \sigma) \cdot f'\|_1$, where $f'$ is the indicator function whose support is $T_\delta(\ell') \cap M_j'$ for some $M_j'$, and $\ell' \in B'_{\mathbf{e}_\Delta, x}$ is the image under $\Gamma$ of the bar $\ell \in B_{\mathbf{e}_\Delta, x}$ corresponding to the $i$th coordinate of $\hat{V}$, i.e., $\ell' = F_\Gamma(\ell)$ where $[\hat{V}]_i = \|\mathcal{N}(x, \sigma) \cdot f\|_1$ and $f$ is the indicator function whose support is $T_\delta(\ell) \cap M_{i_0}$ for some $M_{i_0}$. In other words, $\hat{V}_1'$ is the (not necessarily sorted) vector of weights computed on the bars of $B'_{\mathbf{e}_\Delta, x}$ that are images (under the partial matching $\Gamma$ achieving the bottleneck distance) of the bars of $B_{\mathbf{e}_\Delta, x}$ that were used to generate the (sorted) vector $\hat{V}$.

- the vector $\hat{V}_2'$ of dimension $(N' - N)$ whose $j$th coordinate is $\|\mathcal{N}(x, \sigma) \cdot f'\|_1$, where $f'$ is an indicator function whose support is $T_\delta(\ell') \cap M_j'$ for some $M_j'$, and $\ell' \in B'_{\mathbf{e}_\Delta, x}$ satisfies $\ell' \notin \mathrm{im}(F_\Gamma)$. In other words, $\hat{V}_2'$ is the vector of weights computed on the bars of $B'_{\mathbf{e}_\Delta, x}$ (in an arbitrary order) that are not images of bars of $B_{\mathbf{e}_\Delta, x}$ under $\Gamma$.

Finally, we let $V$ be the vector of dimension $N'$ obtained by filling $\hat{V}$ (whose dimension is $N \leq N'$) with null values until its dimension becomes $N'$, and we let $V' = \mathrm{sort}(\hat{V}')$ be the vector obtained after sorting the coordinates of $\hat{V}'$. Observe that:

$$|V_{\delta,\sigma}(\mathbb{M})(x) - V_{\delta,\sigma}(\mathbb{M}')(x)| = [V - V']_1 = [V - \mathrm{sort}(\hat{V}')]_1 \tag{13}$$

**An upper bound.** We now upper bound $\left\|V - \hat{V}'\right\|_\infty$. Let $q \in [\![1, N']\!]$. Then, one has $[V]_q = \|\mathcal{N}(x, \sigma) \cdot f\|_1$, where $f$ is an indicator function whose support is $T_\delta(\ell) \cap M_i$ for some $M_i$ with $\ell \in B_{\mathbf{e}_\Delta, x}$ if $q \leq N$ and $\ell = \emptyset$ otherwise; and similarly $[\hat{V}']_q = \|\mathcal{N}(x, \sigma) \cdot f'\|_1$, where $f'$ is an indicator function whose support is $T_\delta(\ell') \cap M_j'$ for some $M_j'$ with $\ell' \in B'_{\mathbf{e}_\Delta, x}$. Thus, one has:

$$\begin{aligned}
[V - \hat{V}']_q &= |\; \|\mathcal{N}(x, \sigma) \cdot f\|_1 - \|\mathcal{N}(x, \sigma) \cdot f'\|_1 \;| \\
&\leq \|\mathcal{N}(x, \sigma) \cdot f - \mathcal{N}(x, \sigma) \cdot f'\|_1 \text{ by the reverse triangle inequality} \\
&= \|\mathcal{N}(x, \sigma) \cdot (f - f')\|_1 \text{ by linearity} \\
&\leq \|\mathcal{N}(x, \sigma)\|_2 \cdot \|f - f'\|_2 \text{ by Hölder's inequality} \\
&= (\sqrt{\pi}\sigma)^k \cdot \|f - f'\|_2
\end{aligned}$$

Since $(f - f')$ is an indicator function whose support is $(T_\delta(\ell) \cap M_i) \triangle (T_\delta(\ell') \cap M'_j)$, one has $\|f - f'\|_2 = \sqrt{|(T_\delta(\ell) \cap M_i) \triangle (T_\delta(\ell') \cap M'_j)|}$. Given a segment $\ell$ and a vector $\mathbf{u}$, we let $\ell_\mathbf{u}$ denote the segment $\mathbf{u} + \ell$, and we let $\ell^\mathbf{u}$ denote the (infinite) line induced by $\ell_\mathbf{u}$. More precisely:

$$
\begin{aligned}
\|f - f'\|_2^2 &= |(T_\delta(\ell) \cap M_i) \triangle (T_\delta(\ell') \cap M'_j)| \\
&= |\bigcup_\mathbf{u} (\ell^\mathbf{u} \cap M_i) \triangle \bigcup_\mathbf{u} ((\ell')^\mathbf{u} \cap M'_j)| \\
&\qquad \text{where } \mathbf{u} \text{ ranges over the vectors such that } \|\mathbf{u}\|_\infty \le \delta, \langle \mathbf{u}, \mathbf{e}_\triangle \rangle = 0 \\
&\le \int_\mathbf{u} |(\ell^\mathbf{u} \cap M_i) \triangle ((\ell')^\mathbf{u} \cap M'_j)| \mathrm{d}\mathbf{u} \\
&\le \int_\mathbf{u} |(\ell^\mathbf{u} \cap M_i) \triangle (\ell \cap M_i)_\mathbf{u}| + |(\ell \cap M_i)_\mathbf{u} \triangle (\ell' \cap M'_j)_\mathbf{u}| + |(\ell' \cap M'_j)_\mathbf{u} \triangle ((\ell')^\mathbf{u} \cap M'_j)| \mathrm{d}\mathbf{u} \\
&\le \int_\mathbf{u} 4d_\mathrm{B}(B_{\mathbf{e}_\triangle, x}, B_{\mathbf{e}_\triangle, x+\mathbf{u}}) + 4d_\mathrm{B}(B_{\mathbf{e}_\triangle, x}, B'_{\mathbf{e}_\triangle, x}) + 4d_\mathrm{B}(B'_{\mathbf{e}_\triangle, x}, B'_{\mathbf{e}_\triangle, x+\mathbf{u}}) \mathrm{d}\mathbf{u} \qquad (14) \\
&\le 4 \int_\mathbf{u} \|\mathbf{u}\|_\infty + d_\mathrm{I}(\mathbb{M}, \mathbb{M}') + \|\mathbf{u}\|_\infty \, \mathrm{d}\mathbf{u} \text{ by Lemma A.2 and A.3}
\end{aligned}
$$

Inequality (14) comes from the fact that the symmetric difference between two bars (in two different barcodes) that are both matched (or unmatched) by the optimal partial matching is upper bounded by four times the bottleneck distance between the barcodes, and that (by assumption) the partial matchings achieving $d_\mathrm{B}(B_{\mathbf{e}_\triangle, x}, B_{\mathbf{e}_\triangle, x+\mathbf{u}})$ and $d_\mathrm{B}(B'_{\mathbf{e}_\triangle, x}, B'_{\mathbf{e}_\triangle, x+\mathbf{u}})$ are induced by $\mathbb{M}$ and $\mathbb{M}'$.

**Conclusion.** Finally, one has

$$
\begin{aligned}
|V_{\delta,\sigma}(\mathbb{M})(x) - V_{\delta,\sigma}(\mathbb{M}')(x)| &= [V - V']_1 = [V - \mathrm{sort}(\hat{V}')]_1 \text{ from Equation (13)} \\
&\le \|V - V'\|_\infty \\
&\le (\sqrt{\pi}\sigma)^k \cdot \sqrt{2^{n+1}\delta^{n-1} d_\mathrm{I}(\mathbb{M}, \mathbb{M}') + C_n(\delta)}, \qquad (15)
\end{aligned}
$$

with $C_n(\delta) = 8 \int_\mathbf{u} \|\mathbf{u}\|_\infty \, \mathrm{d}\mathbf{u} \to 0$ when $\delta \to 0$. Inequality (15) comes from the fact that any upper bound for the norm of the difference between a given vector $\hat{V}'$ and a sorted vector $V$, is also an upper bound for the norm of the difference between the sorted version $V'$ of $\hat{V}'$ and the same vector $V$ (see Lemma 3.9 in (Carrière et al., 2015b)). $\qquad \square$

While the stability constant is not upper bounded by $\delta$, $V_{\delta,\sigma}$ is more difficult to compute than the GS representations presented in Definition 3.3.

## C  PROOF OF THEOREM 3.4

Our proof is based on several lemmas. In the first one, we focus on the GS weight function $w$ as defined in Definition 3.3.

**Lemma C.1.** *Let $M$ and $M'$ be two indicator modules with compact support. Then, one has*

$$
d_\mathrm{I}(M, 0) = \frac{1}{2} \sup_{b,d \in \mathrm{supp}(M)} \min_j (d_j - b_j)_+ = w(M). \qquad (16)
$$

*Furthermore, one has the equality*

$$
|w(M) - w(M')| \le d_\mathrm{I}(M, M'). \qquad (17)
$$

*Proof.* We first show Equation (16) with two inequalities.

**First inequality:** $\le$ Let $M$ be an indicator module. If $d_\mathrm{I}(M, 0) = 0$, then the inequality is trivial, so we now assume that $d_\mathrm{I}(M, 0) > 0$. Let $\delta > 0$ such that $\delta < d_\mathrm{I}(M, 0)$. By definition of $d_\mathrm{I}$, the identity morphism $M \to M[2\delta]$ cannot be factorized by $0$. This implies the existence of some $b \in \mathbb{R}^n$ such that $\mathrm{rank}(M(b) \to M(b + 2\delta)) > 0$; in particular, $b, b + 2\delta \in \mathrm{supp}(M)$. Making $\delta$ converge to $d_\mathrm{I}(M, 0)$ yields the desired inequality.

**Second inequality:** $\geq$ Let $(K_n)_{n\in\mathbb{N}}$ be a compact interval exhaustion of $\mathrm{supp}\,(M)$, and $b_n, d_n \in K_n$ be two points that achieve the maximum in

$$\frac{1}{2} \sup_{b,d\in K_n} \min_j (d_j - b_j)_+ .$$

Now, by functoriality of persistence modules, we can assume without loss of generality that $b_n$ and $d_n$ are on the same diagonal line (indeed, if they are not, it is possible to transform $d_n$ into $\tilde{d}_n$ such that $b_n$ and $\tilde{d}_n$ are on the same diagonal line and also achieve the maximum). Thus, $\mathrm{rank}(M(b_n) \to M(d_n)) > 0$, and $d_\mathrm{I}(M,0) \geq \frac{1}{2} \|d_n - b_n\|_\infty$. Taking the limit over $n \in \mathbb{N}$ leads to the desired inequality.

Inequality (17) follows directly from the triangle inequality applied on $d_\mathrm{I}$.

$\square$

In the following lemma, we rewrite volumes of indicator module supports using interleaving distances.

**Lemma C.2.** *Let $M$ be an indicator module, and $R \subseteq \mathbb{R}^n$ be a compact rectangle, with $n \geq 2$. Then, one has:*

$$\mathrm{vol}\,(\mathrm{supp}\,(M) \cap R) = 2 \int_{\{y\in\mathbb{R}^n:y_n=0\}} d_\mathrm{I}\left(M|_{l_y\cap R},0\right)\,\mathrm{d}\lambda^{n-1}(y)$$

*where $l_y$ is the diagonal line crossing $y$, and $\lambda^{n-1}$ denotes the Lebesgue measure in $\mathbb{R}^{n-1}$.*

*Proof.* Using the change of variables $y_i = x_i - x_n$ and $t = x_n$ (which has a trivial Jacobian) yields the following inequalities:

$$\begin{aligned}
\mathrm{vol}(\mathrm{supp}\,(M) \cap R) &= \int_{\mathrm{supp}(M)\cap R} \mathrm{d}\lambda^n(x) \\
&= \int_{\{y\in\mathbb{R}^n:y_n=0\}} \int_{t\in\mathbb{R}} \mathbb{1}_{\mathrm{supp}(M)\cap R}(y+\boldsymbol{t})\,\mathrm{d}t\,\mathrm{d}\lambda^{n-1}(y) \\
&= \int_{\{y\in\mathbb{R}^n:y_n=0\}} \mathrm{diam}_{\|\cdot\|_\infty}\,(\mathrm{supp}\,(M)\cap l_y \cap R)\,\mathrm{d}\lambda^{n-1}(y)
\end{aligned}$$

where $l_y$ is the diagonal line passing through $y$. Now, since $M$ is an indicator module, one has $\mathrm{diam}_{\|\cdot\|_\infty}\,(\mathrm{supp}\,(M)\cap l_y \cap R) = 2d_\mathrm{I}(M|_{l_y\cap R},0)$, which concludes the proof.

$\square$

In the following proposition, we provide stability bounds for single indicator modules.

**Proposition C.3.** *If $M$ and $M'$ are two indicator modules, then for any $\delta > 0$ and GS parameter $\phi_\delta$ in Definition 3.3, one has:*

1. *$0 \leq \phi_\delta(M)(x) \leq \frac{w(M)}{\delta} \wedge 1$, for any $x \in \mathbb{R}^n$,*

2. *$\|\phi_\delta(M) - \phi_\delta(M')\|_\infty \leq 2(d_\mathrm{I}(M,M') \wedge \delta)/\delta$.*

*Proof.* Claim 1. is a simple consequence of Equation (16).

Claim 2. for GS parameter (a) is a simple consequence of the triangle inequality.

Let us prove Claim 2. for (b). Let $x \in \mathbb{R}^n$ and $\delta > 0$. One has:

$$\begin{aligned}
|\phi_\delta(M)(x) - \phi_\delta(M')(x)| &\leq \frac{2}{(2\delta)^n} \int_{\{y:y_n=0\}} \left|d_\mathrm{I}\left(M|_{l_y\cap R_{x,\delta}},0\right) - d_\mathrm{I}\left(M'|_{l_y\cap R_{x,\delta}},0\right)\right|\,\mathrm{d}\lambda^{n-1}(y) \\
&\leq \frac{2}{(2\delta)^n} \int_{\{y:y_n=0\}} d_\mathrm{I}\left(M|_{l_y\cap R_{x,\delta}},M'|_{l_y\cap R_{x,\delta}}\right)\,\mathrm{d}\lambda^{n-1}(y) \\
&\leq 2(d_\mathrm{I}(M,M') \wedge \delta)/\delta,
\end{aligned}$$

where the first inequality comes from Lemma C.2, the second inequality is an application of the triangle inequality, and the third inequality comes from Lemma A.2.

Finally, let us prove Claim 2. for (c). Let $x \in \mathbb{R}^n$ and $\delta > 0$. Let $b \leq d \in \text{supp}\,(M) \cap R_{x,\boldsymbol{\delta}}$. Let also $\gamma > 0$. Then, using Lemma C.2, one has:

$$\frac{1}{(2\delta)^n}\text{vol}(\text{supp}\,(M) \cap R_{b,d}) = \frac{2}{(2\delta)^n} \int_{\{y\in\mathbb{R}^n:y_n=0\}} d_\text{I}(M\big|_{R_{b,d}\cap l_y},0)\,\mathrm{d}\lambda^{n-1}(y)$$

$$\leq \frac{2}{\delta}\gamma + \frac{2}{(2\delta)^n} \int_{\{y\in\mathbb{R}^n:y_n=0\}} d_\text{I}(M\big|_{R_{b+\gamma,d-\gamma}\cap l_y},0)\,\mathrm{d}\lambda^{n-1}(y),$$

using the convention $R_{a,b} = \varnothing$ if $a \not\leq b$. Now, set $\gamma := d_\text{I}(M\big|_{R_{x,\boldsymbol{\delta}}}, M'\big|_{R_{x,\boldsymbol{\delta}}})$. If $b + \boldsymbol{\gamma}$ or $d - \boldsymbol{\gamma} \notin \text{supp}\,(M')$ then $d_\text{I}(M\big|_{R_{x,\boldsymbol{\delta}}}, M'\big|_{R_{x,\boldsymbol{\delta}}}) = \gamma > d_\text{I}(M, M')$ which is impossible. Thus,

$$\frac{1}{(2\delta)^n}\text{vol}(R_{b,d}) \leq 2d_\text{I}(M\big|_{R_{x,\boldsymbol{\delta}}}, M'\big|_{R_{x,\boldsymbol{\delta}}})/\delta$$

$$+ \sup_{a,c\in R_{x,\boldsymbol{\delta}}\cap\text{supp}(M')} \frac{2}{(2\delta)^n} \int_{\{y\in\mathbb{R}^n:y_n=0\}} d_\text{I}(M\big|_{R_{a,c}\cap l_y},0)\,\mathrm{d}\lambda^{n-1}(y)$$

$$= 2d_\text{I}(M\big|_{R_{x,\boldsymbol{\delta}}}, M'\big|_{R_{x,\boldsymbol{\delta}}})/\delta + \phi_\delta(M')(x)$$

Finally, taking the supremum on $b \leq d \in \text{supp}\,(M) \cap R_{x,\boldsymbol{\delta}}$ yields

$$\phi_\delta(M)(x) - \phi_\delta(M')(x) \leq 2d_\text{I}(M\big|_{R_{x,\boldsymbol{\delta}}}, M'\big|_{R_{x,\boldsymbol{\delta}}})/\delta \leq 2\left(d_\text{I}(M, M') \wedge \delta\right)/\delta.$$

The desired inequality follows by symmetry on $M$ and $M'$.

$\square$

Equipped with these results, we can finally prove Theorem 3.4.

*Proof. Theorem 3.4.*

Let $\mathbb{M} = \oplus_{i=1}^m M_i$ and $\mathbb{M}' = \oplus_{j=1}^{m'} M_j'$ be two modules that are decomposable into indicator modules and $x \in \mathbb{R}^n$.

**Inequality 5.** To simplify notations, we define the following: $w_i := w(M_i)$, $\phi_{i,x} := \phi_\delta(M_i)(x)$ and $w_j' := w(M_j')$, $\phi_{j,x}' := \phi_\delta(x, M_j')$. Let us also assume without loss of generality that the indices are consistent with a matching achieving the bottleneck distance. In other words, the bottleneck distance is achieved for a matching that matches $M_i$ with $M_i'$ for every $i$ (up to adding 0 modules in the decompositions of $\mathbb{M}$ and $\mathbb{M}'$ so that $m = m'$). Finally, assume without loss of generality that $\sum_i w_i' \geq \sum_i w_i$. Then, one has:

$$|V_{1,\delta}(\mathbb{M})(x) - V_{1,\delta}(\mathbb{M}')(x)| = \left|\frac{1}{\sum_i w_i}\sum_i w_i\phi_{i,x} - \frac{1}{\sum_i w_i'}\sum_i w_i'\phi_{i,x}'\right|$$

$$\leq \frac{1}{\sum_i w_i'}\left|\sum_i w_i\phi_{i,x} - \sum_i w_i'\phi_{i,x}'\right| + \left|\frac{1}{\sum_i w_i} - \frac{1}{\sum_i w_i'}\right|\left|\sum_i w_i\phi_{i,x}\right|.$$

Now, for any index $i$, since $d_\text{I}(M_i, M_i') \leq d_\text{B}(\mathbb{M}, \mathbb{M}')$ and $|w_i - w_i'| \leq d_\text{I}(M_i, M_i') \leq d_\text{B}(\mathbb{M}, \mathbb{M}')$ by Lemma C.1, Proposition C.3 ensures that:

$$|w_i\phi_{i,x} - w_i'\phi_{i,x}'| \leq |w_i - w_i'|\phi_{i,x} + w_i'|\phi_{i,x} - \phi_{i,x}'| \leq 2(w_i + w_i')(d_\text{B}(\mathbb{M}, \mathbb{M}') \wedge \delta)/\delta$$

and

$$\left|\frac{1}{\sum_i w_i} - \frac{1}{\sum_i w_i'}\right| \leq \frac{1}{\sum_i w_i'}\left|\frac{\sum_i w_i' - w_i}{\sum_i w_i}\right| \leq \frac{md_\text{B}(\mathbb{M}, \mathbb{M}')}{(\sum_i w_i')(\sum_i w_i)}.$$

Finally,

$$|V_{1,\delta}(\mathbb{M})(x) - V_{1,\delta}(\mathbb{M}')(x)| \leq \left[\frac{\sum_i w_i + w_i'}{\sum_i w_i'} + \frac{\sum_i w_i}{\frac{1}{m}(\sum_i w_i)(\sum_i w_i')}\right]2(d_\text{B}(\mathbb{M}, \mathbb{M}') \wedge \delta)/\delta$$

$$\leq \left[4 + \frac{2}{C}\right](d_\text{B}(\mathbb{M}, \mathbb{M}') \wedge \delta)/\delta.$$

**Inequality 4** can be proved using the proof of Inequality 5 by replacing every $w_i$ by 1.

**Inequality 6.** Let us prove the inequality for (b). Let $R := R_{x-\boldsymbol{\delta}, x+\boldsymbol{\delta}}$. One has:

$$
\begin{aligned}
V_{\infty,\delta}(\mathbb{M})(x) - V_{\infty,\delta}(\mathbb{M}')(x) = \\
\sup_i \frac{2}{(2\delta)^n} \int_{\{y \in \mathbb{R}^n : y_n = 0\}} d_{\mathrm{I}}(M_i\big|_{l_y \cap R}, 0) \, \mathrm{d}\lambda^{n-1}(y) \\
- \sup_j \frac{2}{(2\delta)^n} \int_{\{y \in \mathbb{R}^n : y_n = 0\}} d_I(M_j'\big|_{l_y \cap R}, 0) \, \mathrm{d}\lambda^{n-1}(y) \\
\textit{(for any index } j) \leq \sup_i \frac{2}{(2\delta)^n} \int_{\{y \in \mathbb{R}^n : y_n = 0\}} d_{\mathrm{I}}(M_i\big|_{l_y \cap R}, 0) - d_{\mathrm{I}}(M_j'\big|_{l_y \cap R}, 0) \, \mathrm{d}\lambda^{n-1}(y) \\
\leq \frac{2}{(2\delta)^n} \int_{\{y \in \mathbb{R}^n : y_n = 0\}} \sup_i \inf_j d_{\mathrm{I}}(M_i\big|_{l_y \cap R}, M_j'\big|_{l_y \cap R}) \, \mathrm{d}\lambda^{n-1}(y)
\end{aligned}
$$

Now, as the interleaving distance is equal to the bottleneck distance for single parameter persistence (Chazal et al., 2016, Theorem 5.14), one has:

$$
\sup_i \inf_j d_{\mathrm{I}}(M_i\big|_{l_y \cap R}, M_j'\big|_{l_y \cap R}) \leq d_{\mathrm{B}}(\mathbb{M}\big|_{l_y \cap R}, \mathbb{M}'\big|_{l_y \cap R}) = d_{\mathrm{I}}(\mathbb{M}\big|_{l_y \cap R}, \mathbb{M}'\big|_{l_y \cap R}) \leq d_{\mathrm{I}}(\mathbb{M}, \mathbb{M}') \wedge \delta
$$

which leads to the desired inequality. The proofs for (a) and (c) follow the same lines (upper bound the suprema in the right hand term with either infima or appropriate choices in order to reduce to the single parameter case). $\qquad\square$

## D    PSEUDO-CODE FOR OUR REPRESENTATIONS

In this section, we briefly present the pseudo-code that we use to compute our GS representations. Our code is based on implicit descriptions of the decompositions of multiparameter persistence modules (which are the inputs of our GS representations) through their so-called *birth* and *death* corners. These corners can be obtained with, e.g., the public softwares MMA (Loiseaux et al., 2022) and RIVET (Lesnick & Wright, 2022).

In order to compute our GS representations, we implement the following procedures:

1. Given an indicator module $M$ and a rectangle $R \subset \mathbb{R}^n$, compute the restriction $M\big|_R$.

2. Given a module $M$ (or $M\big|_R$), compute $d_{\mathrm{I}}(M, 0)$. This allows (combined with the first algorithm above) to compute our weight function and first summand representation in Definition 3.3.

3. Given a module restricted to a rectangle, compute the volume of the biggest rectangle in the support of this module. This allows (combined with the first algorithm above) to compute the third summand representation in Definition 3.3.

For the first point, Algorithm 1 corresponds to "pushing" the corners of the module on the given rectangle in order to obtain the updated corners.

---

**Algorithm 1:** Restriction of a module to a rectangle

---

**Data:** birth and death corners of a module $M$, rectangle $R = \{z \in \mathbb{R}^n : m \leq z \leq M\}$
**Result:** new_summands_corners, the birth and death corners of $M\big|_R$.
**for** summand $= \{$summand_birth_corners, summand_death_corners$\}$ ***in*** $M$ **do**
    new_birth_list $\leftarrow []$;
    **for** $b$ ***in*** summand_birth_corners **do**
        **if** $b \leq M$ **then**
            $b' = \{\max(b_i, m_i) \text{ for } i \in [\![1, n]\!]\}$;
            Append $b'$ to new_birth_list;
        **end**
    **end**
    new_death_list $\leftarrow []$;
    **for** $d$ ***in*** summand_death_corners **do**
        **if** $d \geq m$ **then**
            $d' = \{\min(d_i, M_i) \text{ for } i \in [\![1, n]\!]\}$;
            Append $d'$ to new_death_list;
        **end**
    **end**
    new_summands_corners $\leftarrow$ [new_birth_list, new_death_list];
**end**

---

For the second point, we proved in Lemma C.1 that our GS weight function is equal to $d_I(M, 0)$ and has a closed-form formula with corners, that we implement in Algorithm 2.

---

**Algorithm 2:** GS weight function

---

**Data:** birth and death corners of a module $M$
**Result:** distance, the interleaving distance $d_I(M, 0)$.
distance $\leftarrow 0$;
**for** $b$ ***in*** $M$_birth_corners **do**
    **for** $d$ ***in*** $M$_death_corners **do**
        distance $\leftarrow \max\left(\text{result}, \frac{1}{2}\min_i(d_i - b_i)_+\right)$;
    **end**
**end**

---

The third point also has a closed-form formula with corners, leading to Algorithm 3.

---

**Algorithm 3:** GS summand representation

---

**Data:** birth and death corners of a module $M$
**Result:** volume, the volume of the biggest rectangle fitting in $\text{supp}(M)$
volume $\leftarrow 0$;
**for** $b$ ***in*** $M$_birth_corners **do**
    **for** $d$ ***in*** $M$_death_corners **do**
        volume $\leftarrow \max\left(\text{result}, \Pi_i(d_i - b_i)_+\right)$;
    **end**
**end**

---

## E   Running time comparisons

In this section, we provide running time comparisons between our GS representations and the MPI and MPL representations from the literature. It is worth noting that our implementation relies on an implicit description of the summands in the module decomposition, which are characterized with their *corners* (Loiseaux et al., 2022). This is in contrast with the implementations of the other representations that are available in the `multipers`[4] library, which require slicing the modules with many lines first. This shows from Table 3, where it can be seen that our representations

---
[4] https://github.com/MathieuCarriere/multipers

(computed on the pinched annulus and immunohistochemistry data sets defined in Section 4) can be computed much faster than the others, by a factor of at least 25 (all representations are evaluated on a grid of size $100 \times 100$, and we provide the maximum running time over $p \in \{0, 1, \infty\}$). All computations were done using a Ryzen 4800 laptop CPU, with 16GB of RAM. Interestingly, this sparse and fast implementation based on corners can also be used to improve on the running time of the multiparameter persistence landscapes (MPL), as one can see from our pseudo-code in Algorithm 4 (which retrieves the persistence barcode of a multiparameter persistence module along a given line; this is enough to compute the MPL) and from the second line of Table 3.

---

**Algorithm 4:** Restriction of a module to a line

---

**Data:** birth and death corners of a module $M$, a diagonal line $l$
**Result:** barcode, the persistence barcode associated to $M\big|_l$
barcode $\leftarrow [\,]$;
$y \leftarrow$ an arbitrary point in $l$;
**for** summand $= \{$summand_birth_corners, summand_death_corners$\}$ ***in*** $M$ **do**
    birth $\leftarrow y + \mathbf{1} \times \min_{b \in \text{summand\_birth\_corners}} \max_i b_i - y_i$;
    death $\leftarrow y + \mathbf{1} \times \max_{d \in \text{summand\_death\_corners}} \min_i d_i - y_i$;
    bar $\leftarrow [\text{birth}, \text{death}]$;
    Append bar to barcode;
**end**

---

|  | Annulus | Immuno |
|---|---|---|
| Ours (GS) | $250ms \pm 2ms$ | $275ms \pm 9.8ms$ |
| Ours (MPL) | $36.9ms \pm 0.8ms$ | $65.9ms \pm 0.9ms$ |
| MPI (50) | $6.43s \pm 25ms$ | $5.67s \pm 23.3ms$ |
| MPL (50) | $17s \pm 39ms$ | $15.6s \pm 14ms$ |
| MPI (100) | $13.1s \pm 125ms$ | $11.65s \pm 7.9ms$ |
| MPL (100) | $35s \pm 193ms$ | $31.3s \pm 23.3ms$ |

Table 3: Running times for our GS representations and competitors.

