# OpenReview forum: "A sparse, fast, and stable representation for multiparameter topological data analysis"
_ICLR.cc/2023/Conference — Submitted to ICLR 2023_

### Official Review · Reviewer_su2T · 2022-10-19

**Confidence:** 2
**Correctness:** 4
**Technical Novelty And Significance:** 3
**Empirical Novelty And Significance:** 3
**Recommendation:** 6

**Clarity, Quality, Novelty And Reproducibility:**

## Clarity

There is an clear effort to adapt the article to a community not familiar with persistent homology. Thus, the paper may sometime be hard to follow but the authors always put a citation for help. Moreover the figure helps to better understand most of the statements.

Still there is little issue with the citation, please use \citep when making reference to paper when the name of the authors in not in the text.

## Quality

The proposed framework is interesting and there is an effort to make the paper readable.

## Novelty

I am not sure on how new is the framework. Both main ingredient exist, but some part rely on recent results. The full framework is clearly new.

## Reproducibility

Even for someone not familiar with persistent homology, the appendix gives the information needed to implement the method.  The code is also available.

**Strength And Weaknesses:**

## Strength

The proposed framework is interesting on several points.
1. The combination of multiparameter persistency with persistent homology allows to catch complex features inside the data. The multiparameter is then crucial to deal with data with more than one dimension.
2. The representation is robust (using geometric and stable parameters) so usable in context where noise may be important.
3. There is an analysis of both the approximation quality of the feature and the convergence toward a complete representation of the data.

## Weaknesses

Such framework have several weakness,
1. High dimensional data are still an issue with such framework.
2. The scope of such methods is unclear and should be discuss.

**Summary Of The Paper:**

This paper presents a new topological data analysis framework based on persistent homology. The main contributions are first the general framework with multiparameter persistency and a sparse implementation. The experiments show interesting results on point cloud classification.

**Summary Of The Review:**

This article proposes a framework for topological data analysis. It combines multiparameter persistency with persistent homology to produce robust feature to describe data. The features are shown to produce a controlled approximation of the data robust to noise. Experiments show interesting results for classification of complex data. This paper is easy to read and may reach a larger public than just its community.

---

> ### Author Response · Authors · 2022-11-18
> **Answer to review**
>
> - "High dimensional data are still an issue with such framework"
>
> The reviewer raises a good question, and we apologize for being unclear on this point.  In fact, one of the virtues of this kind of method is that the efficiency and algorithmic complexity only depend on the size of the data set and the intrinsic dimensionality of the circle, not the dimension of the ambient space.  Under the standard hypothesis that the signal lies on a low-dimensional subspace, these methods are quite practical.
>
> To be more precise, the efficiency of our representation only depends on the complexity of the module: specifically, the running time is approximately linear w.r.t. the number of summands in the module, which is only related to the size of the data and the intrinsic dimension.
>
> - "The scope of such methods is unclear and should be discuss."
>
> Thank you for the comment. We have modified the introduction to better articulate the positioning of this work, and to better highlight our contribution (see general comment).

---

> > ### Comment · Reviewer_su2T · 2022-11-30
> > **New Official Comment**
> >
> > Thanks for your clarifications.

---

### Official Review · Reviewer_WBCA · 2022-10-25

**Confidence:** 1
**Correctness:** 3
**Technical Novelty And Significance:** 3
**Empirical Novelty And Significance:** Not applicable
**Recommendation:** 6

**Clarity, Quality, Novelty And Reproducibility:**

The paper proposes a general framework for multiparameter persistent homology, and parameters that consider geometry and stability. They contribute to the novelty of the paper.

The paper is difficult to read for the audience without a certain background.


**Strength And Weaknesses:**

Strength:
- The proposed framework unifies previous work by selecting different operations and functions. The proposed parameters consider geometry and stability, which are effective in classification.
- Experiments show the proposed method has fast convergence rate, and better quality than others.

Weakness
- The paper has heavy theory, and it’s not easy to read for the audience without a background in topological data analysis and persistent homology.


**Summary Of The Paper:**

The paper proposes a framework to use multiple parameters to compute persistent homology for topological data analysis. The proposed sparse implementation reduces memory usage and running time. Experiments show the proposed method converges fast and achieves the best performance on several classification tasks.

**Summary Of The Review:**

The paper proposes a general framework for multiparameter persistent homology with geometric and stable parameters.

---

> ### Author Response · Authors · 2022-11-18
> **Answer to review**
>
> - "The paper is difficult to read for the audience without a certain background."
>
> Thank you for the comment. We have added more explanations about the steps of our constructions in the main body of the article (see also general comment).

---

### Official Review · Reviewer_Bro2 · 2022-10-25

**Confidence:** 3
**Correctness:** 3
**Technical Novelty And Significance:** 2
**Empirical Novelty And Significance:** 2
**Recommendation:** 5

**Clarity, Quality, Novelty And Reproducibility:**

Clarity: The paper is hard to read due to the heavy math and almost lack of words (I could not had time to read most of the presented Math to check their correctness). The paper should properly balance both math and text. At the beginning of this paper, the authors state that they present an approach for shape modeling, but then, they show some basic results on toy datasets selected from UCR. That does not make a well structured and a complete story to me. The authors can focus on matching the beginning of the paper (their initial motivation) and their presented approach with its results.

Quality: It seems there is some novelty here but the way it is presented, makes it harder to value its significance. The experimental results seem basic, and the reported improved results seem marginal. None of those shows any significance. While there are many multiparametric algorithms out there, the authors choose to compare their technique to more basic approaches.

Novelty: As I stated above, it seems that this paper has some novelty but the main ideas also seem to be borrowed from scale-space theory and from geometry based multi-parameter computing concepts. Both concepts are studied in the literature (see some sample works from the literature above). Consequently, the novelty of this work and its motivation could be emphasized in a more clear way.

Reproducibility: I am not sure how well the presented results could be replicated without the code. Seems like there are many handpicked parameters that are not fully described in Algorithm 1. The authors claim to release their code upon acceptance, however.




**Strength And Weaknesses:**

Strengths:

- The paper introduces a multiparameter based approach for modeling. The authors provide proof for their theorem (which I did not check for correctness).

- The run time of the algorithm seems faster when compared to the run time of the used baseline algorithms as shown in Table 2.

- The algorithm shows more than 10 % improvement on accuracy, when compared to MPL and Ripley. (However, more baseline algorithms could also be used there for a better judgment).


Weaknesses:
- The idea of using multiple parameters for geometric modeling and for filtering is not new, and there are other multiparameter based solutions exist in the literature. The authors can do a better taxonomy of those works from the literature by including them in their work and in their experiments (for example, the authors should also include the following 3 works as they also are highly related to this work: (i) O. Chapelle, V. Vapnik, O. Bousquet, and S. Mukherjee, "Choosing multiple parameters for support vector machines", Machine learning, 46(1-3):131–159, 2002; and (ii) Ozer, S. (2018), "Similarity domains machine for scale-invariant and sparse shape modeling", IEEE Transactions on Image Processing, 28(2), 534-545; (iii) Witkin, Andrew P. "Scale-space filtering." In Readings in Computer Vision, pp. 329-332. Morgan Kaufmann, 1987)

- The paper uses heavy Math based representation, which is hard to follow. Furthermore, the proposed "algorithm" is not clear to understand. The structure of this paper could be improved by focusing the flow of the paper to explain the steps of the proposed approach first. Currently the proposed algorithm is given as Appendix and it is not citing the any particular equation from the main text. That could be simplified.

- Experimental results show marjinal improvement on toy datasets and the approach is compared to k-NN classifier along with 3 additional random algorithms in Table 3.  The baseline algorithms seem limited. For classification, the authors could use many classification algorithms, when their performance metric is accuracy.

In Table 3, the caption states that "classification results for time series". It is not well explained how the time series problem is considered as a geometry based multi-scale classification problem there. That could be clarified.


**Summary Of The Paper:**

This paper mentions introduces a multiparameter persistent homology based solution to the "finding shape descriptors problem" in point clouds.  The authors also introduce their solution for topological data analysis as a generic tool. The authors introduce some upper limits for the stability for their work.



**Summary Of The Review:**

This paper mentions introduces a multiparameter persistent homology based solution to the "finding shape descriptors problem" in point clouds.  The authors also introduce their solution for topological data analysis as a generic tool. The authors introduce some upper limits for the stability for their work.

Overall, due to the vast amount of Math usage, instead of words, this paper remains hard to read and requires way more time to understand fully (as a reviewer, I could not give that much time to check the correctness of each equation, unfortunately). The used ideas could be found in the literature. As such, the motivation and the novelty of this paper remains unclear to me. The experimental results include only a few baseline algorithms and they run on toy datasets. There are a vast amount of classifiers out there, and the authors could include a large amount of classifiers to support their claims in their experiments.

---

> ### Author Response · Authors · 2022-11-18
> **Answer to review**
>
> - "The idea of using multiple parameters for geometric modeling and for filtering is not new, and there are other multiparameter based solutions exist in the literature. The authors can do a better taxonomy of those works from the literature by including them in their work and in their experiments"
>
> Thank you for these suggestions. We have added these references in the article.  We note that we do not claim any novelty for the idea of using multiparameter approaches in geometric modeling.  We believe that the motivations for using multiparameter topology (as opposed to the approaches that appear in the literature derived from the references the referee mentions) have been amply described in the literature, and while we want to indicate the broader universe of antecedents our focus is not on comparing topological methods vs non-topological ones.
>
> - "The paper uses heavy Math based representation, which is hard to follow. Furthermore, the proposed "algorithm" is not clear to understand. The structure of this paper could be improved by focusing the flow of the paper to explain the steps of the proposed approach first. Currently the proposed algorithm is given as Appendix and it is not citing the any particular equation from the main text. That could be simplified."
>
> Thank you for this comment. We have added more explanations in the appendix that help relating our pseudo-code with the actual definition
> of our representation, and we have added more text in the main body of the article that describes the steps of our constructions more precisely.
>
> - "Experimental results show marginal improvement on toy datasets and the approach is compared to k-NN classifier along with 3 additional random algorithms in Table 3. The baseline algorithms seem limited. For classification, the authors could use many classification algorithms, when their performance metric is accuracy."
>
> Our goal with the evaluation was to highlight the performance of our algorithm in light of the fact that (a) topological methods are comparable to existing techniques on standard benchmarks, and (b) there is a class of examples (e.g., stereochemistry data sets) where geometric approaches such as this are clearly superior.  We take seriously the reviewers criticism that a more detailed comparison would be helpful.
>
> In order to improve the non-topological baselines, we have added the performances obtained from a few additional classifiers (SVM, XGBoost, Random Forests) trained on the Ripley function. We have also extended and added some of the time series result in the main body of the article to better emphasize the generality of our approach. Baselines for both of these real-world datasets (immunohistochemistry and time series) were chosen based on standard practices in these fields (see for instance the articles https://www.medrxiv.org/content/10.1101/2021.04.27.21256104v1.full.pdf and https://arxiv.org/pdf/1810.07758.pdf).
>
> The conclusion that we intend to highlight is that our method is comparable to other topological methods (and to the state of the art generic classification algorithms) on the toy benchmark examples but more interestingly is the best on representative data sets that are known to have geometric structure.
>
> - "In Table 3, the caption states that "classification results for time series". It is not well explained how the time series problem is considered as a geometry based multi-scale classification problem there. That could be clarified."
>
> The procedure we follow to classify time series geometrically is through time-delay embedding, which is a standard approach.  In the specific context we work with, this is for example reviewed in reference [Multiparameter persistence images for topological machine learning, Carrière, Blumberg, NeurIPS, 2020]. We have added details about this in the text; we apologize for being too brief in our original discussion of this.
>
> The basic idea of time-delay embedding is to turn every measurement of a time series into a point in Euclidean space R^n by using the measurement value and the (n-1) following ones as coordinates. This allows to turn any time series into a point cloud, on which we can apply our topological tools. Moreover, some properties of the time series translates directly into geometric properties of the associated point cloud: for instance, periodicity usually creates loops in the point clouds, that can then be detected with topology.

---

> > ### Comment · Reviewer_Bro2 · 2022-11-24
> > **Additional data, the proper definition of the used terms and the assumptions made, are not all made clearly.**
> >
> > I would like to thank the authors for providing a feedback. While this version seems improved, in its current form, It still lacks a satisfactory response for my main concerns and questions. It also still lacks providing a proper relation between the existing work and this "new" work in terms of shape representation. I believe stating generic and bold claims such as: "providing a general framework" requires first a more comprehensive and thorough (in depth) set of definitions and assumptions. A shape descriptor (such as the famous SIFT descriptor) requires a proper relation to a shape. Nowever, I do not see a clear relation to a shape in this paper. It makes me think what the authors mean with the term "shape". Therefore, while the authors provide some feedback, they still lack answering my such concerns. In this form, the paper reads like a confusing paper, that tries to combine multiple terms in a single paper including multiparameter modeling, shape representation, graph modeling, filtering, etc. However there is still no clear explanation what the authors mean with each of those terms as they use in this paper. That creates a significant amount of confusion and makes the paper read like a mixture of various terms. That needs to be fixed.
> >
> > Some additional points:
> >
> > - The authors state that "In order to improve the non-topological baselines, we have added the performances obtained from a few additional classifiers (SVM, XGBoost, Random Forests) trained on the Ripley function". However, I could not see those additional results in the tables. I suggest them to check that.
> >
> > - The references that I randomly selected and provided in my review were just to give samples to the authors so that they could provide the differences and similarities  between those references and this work by elaborating on them. All those references mention about either multiple-parameter modeling or scale invariant shape representation. Currently, the authors just add those references with a single sentence. However, there is no discussion on how and in which ways their work is similar to, and different from those references. Such a discussion would clarify the distinction between those works and this paper. Maybe a section or a subsection on such discussion, describing why and how this work differs from such relevant literature, would increase the value of this paper while increasing its clarity. I suggest adding such a discussion which would also help clarifying the motivation of this paper.
> >
> > - There are still many terms that are not properly defined in this paper. In this kind of a math-heavy paper, each and every single term should be defined before being used and all the assumptions should be provided at the beginning. Currently that is not the case. Some variables are associated with some words but it is not clear what the authors mean with each and every single one of those words/terms (different words have different meanings in different domains). I suggest authors to focus on providing such definitions and assumptions clearly first.
> >
> > - I also thank the authors for clarifying their approach on using time varying dataset. However, that approach works only on stationary datasets and I suggest them to mention about such limitation in their paper as well.
> >
> > - The authors can also add some references for their claim: "A standard procedure for studying a data set provided as a finite metric space, i.e., a set X with a distance dX : X × X → R, is to form a graph with vertex set X and an each edge connecting xi and xj when dX (xi , xj ) < ε."
> >
> > - Figures (including Fig. 1) are not all clear to me. It is not clear what we should understand from each figure and the authors should revise those figure captions to guide the reader on that matter.
> >
> > Overall, while I think the paper might have some merits, its presentation might require a thorough revision to clarify all the points, terms and claims made in this paper and  needs improvements on its experimental results section with more compelling results.

---

### Official Review · Reviewer_a971 · 2022-12-05

**Confidence:** 4
**Correctness:** 3
**Technical Novelty And Significance:** 4
**Empirical Novelty And Significance:** 4
**Recommendation:** 5

**Clarity, Quality, Novelty And Reproducibility:**

- The paper is generally well-written, but some parts should be improved for clarity.  See Weakness.
- The paper contains sufficiently novel contributions to multiparameter persistent modules in topological data analysis.


**Strength And Weaknesses:**

Strength:
- A framework for representing multiparameter persistence modules is proposed, covering some previous methods.
- The proposed methods with specific representations have a theoretical guarantee of stability, which is an important property of a topological feature in applications.
- In the two multiparameter cases of the measure bifiltration and Cech complex with density, the theoretical analysis gives convergence rates of the proposed GS representations, and the rates are also confirmed numerically.
- The experimental results with real-world data sets demonstrate better classification accuracies than the existing multiparameter representations.

Weakness:
- The significance of the proposed framework and method is not clearly presented in the paper.
    - While the multiparameter persistence is a more general approach, the practical advantage in data analysis over 1-parameter persistence is not necessarily clear in the paper.  For the real-world data, more basic comparisons with 1-parameter persistent homology as well as other relevant methods should be included.
    - The advantage over the existing multiparameter methods is not sufficiently explained, although it is somehow discussed.  One of the advantages should be stability.  In comparison with Carriere & Blumberg (2022),  the authors just cite Botnan & Lesnick (2018) without details.  As an important contribution, the reason for having stability, unlike the others, should be explained more clearly and demonstrated with some basic numerical examples.
- The significance of the theoretical analysis in Section 4.1 is unclear.
    - As discussed in the paper, the dependence of constant $\delta$ is not clear enough.  While the authors say that the empirical results do not show bad behavior by choice of \delta, more detailed discussions are necessary about whether this dependence on $\delta$ appears as an artifact by the theoretical derivation or an inevitable factor by the method.
    - In the upper bound (8), the bandwidth parameter $h$ remains constant, and the bound involves a constant factor.  Can we say that the first term $1/\sqrt{n}$ is the convergence rate?
- Some parts of the paper are tough to understand for standard readers in the machine learning field.  Some examples are:
    - In the last paragraph of page 3, "indecomposable" module is undefined.
    - In Figure 1, it is not clearly explained what is depicted in the middle and right figures.  More explanations are needed.
- The paper may not be easy to read for most researchers in the machine learning field.   While I understand the difficulty of explaining persistent homology to non-experts, the authors could demonstrate better the importance of the problem to be solved in this paper.  Also, I recommend the authors include a brief introduction about the basic notions of persistence homology in Supplementary Materials.


**Summary Of The Paper:**

This paper discusses a framework of multiparameter persistence, which has been a long-standing problem in the field of topological data analysis with persistence homology, and proposes three types of specific representations (GS representations) with stability guarantee.   The numerical results demonstrate the convergence rates with respect to the sample size and the effectiveness in classification accuracies with real-world data.


**Summary Of The Review:**

I think that the paper contains significant contributions to the problem of multiparameter persistent modules, which is one of the important issues in topological data analysis.  However, the presentation of the paper has much room for improvement, as detailed in Strength and Weaknesses.

---

### Author Response · Authors · 2022-11-18
**General answer**

We first want to thank the reviewers for their comments.

All reviewers pointed out the fact that the scope and contributions of this work are unclear, which is an obvious issue with our writing. We have added a paragraph in the introduction to better position and articulate our contribution.

We want to begin by explaining our contribution in this general comment. Our contribution belongs to the field of multiparameter topological data analysis (TDA), which is both highly promising for problems with multiple natural scale parameters and has been hampered by difficulties in balancing theoretical guarantees with reasonable performance.

Our contribution lies in the design of a novel multiparameter TDA representation that is both the most expressive in the literature and also comes with new theoretical guarantees that are unavailable for other approaches.  It is of course the case that the idea of using multiparameter representations or optimizing parameters has been around for a long time, and we do not claim novelty for this idea per se.

We now describe the contributions a bit more precisely. While a few other representations for multiparameter TDA already exist, ours is, to our knowledge, the first and only one to be both rich---in terms of how much information it encodes (which we show in the paper by defining it from decompositions that are stronger quantities than the so-called rank invariant) and provably robust to noise at the same time (which we show in the paper by proving a stability result, that directly leads to theoretical convergence rates). Current methods in the state-of-the-art are either much less expressive or are known to be not robust.

Then, we designed our experiments to show that as a sanity check our method (like all topological methods) is in the ballpark for standard benchmarks and is superior for a class of problems that TDA methods are known to outperform standard methods on.  Specifically, the scores of our representations compared remarkably well against other TDA representations, for both accuracy and running times on synthetic and real-world datasets.

---

### Decision · Program_Chairs · 2023-01-20

**Decision:**

Reject

**Justification For Why Not Higher Score:**

However, as pointed out by reviewers, the paper has some clarity issues. For example, the significance of the proposed framework and method is not clearly presented. Thus, it needs a major revision before publication.

**Justification For Why Not Lower Score:**

N/A

**Metareview: Summary, Strengths And Weaknesses:**

In this paper, the authors propose a framework of multiparameter persistence, which has been a long-standing problem in the field of topological data analysis with persistence homology.  Then, the authors introduced their solution for topological data analysis as a generic tool, and demonstrated that the proposed framework is useful in practice. However, as pointed out by reviewers, the paper has some clarity issues. For example, the significance of the proposed framework and method is not clearly presented. Thus, it needs a major revision before publication.

I encourage the authors to update the paper based on the reviewer's comments and resubmit it to a future venue.



**Summary Of Ac-Reviewer Meeting:**

n/a